

# An intelligent diabetes classification and perception framework based on ensemble and deep learning method

Qazi Waqas Khan[1], Khalid Iqbal[2], Rashid Ahmad[2,3], Atif Rizwan[1], Anam Nawaz Khan[1] and DoHyeun Kim[1]

[1] Department of Computer Engineering, Jeju National University, South Korea, Jeju-si, Jeju, South Korea
[2] Department of Computer Science, COMSATS University Islamabad, Attock Campus, Attock, Punjab, Pakistan
[3] Bigdata Research Center, Jeju National University, Jeju-si, Jeju, South Korea

## ABSTRACT

Sugar in the blood can harm individuals and their vital organs, potentially leading to blindness, renal illness, as well as kidney and heart diseases. Globally, diabetic patients face an average annual mortality rate of 38%. This study employs Chi-square, mutual information, and sequential feature selection (SFS) to choose features for training multiple classifiers. These classifiers include an artificial neural network (ANN), a random forest (RF), a gradient boosting (GB) algorithm, Tab-Net, and a support vector machine (SVM). The goal is to predict the onset of diabetes at an earlier age. The classifier, developed based on the selected features, aims to enable early diagnosis of diabetes. The PIMA and early-risk diabetes datasets serve as test subjects for the developed system. The feature selection technique is then applied to focus on the most important and relevant features for model training. The experiment findings conclude that the ANN exhibited a spectacular performance in terms of accuracy on the PIMA dataset, achieving a remarkable accuracy rate of 99.35%. The second experiment, conducted on the early diabetes risk dataset using selected features, revealed that RF achieved an accuracy of 99.36%. Based on our experimental results, it can be concluded that our suggested method significantly outperformed baseline machine learning algorithms already employed for diabetes prediction on both datasets.

## INTRODUCTION

High blood glucose levels stand as a primary cause of death, rendering diabetes a devastating chronic condition. From 1980 to 2014, the number of individuals grappling with diabetes surged dramatically, from 108 million to 422 million, as reported by the World Health Organization. In the adult population, diabetes affects 8.5%. Moreover, it impacts 30.3% of the U.S. population (*World Health Organization, 2016*). Among the most populous nations, China and India boast some of the highest diabetes rates—98 million and 65.1 million, respectively (*Todkar, 2016*). Both type 1 and type 2 diabetes are severe conditions. Type 1 diabetes involves the targeted destruction of insulin-producing

Corresponding author
DoHyeun Kim, kimdh@jejunu.ac.kr

pancreatic cells, while insulin resistance underlies type 2 diabetes (*Raha et al., 2009*; *Bellamy et al., 2009*). Although there is currently no cure for diabetes, early detection can significantly reduce the likelihood of complications. A well-balanced diet and prompt diagnosis contribute to enhancing an individual's lifespan. Early diabetes detection based on a doctor's subjective judgment can be inaccurate due to gaps in our understanding of the associated patterns (*Palaniappan & Awang, 2008*). Predictive analytics offers the potential to identify those at risk for diabetes better, anticipate issues, and improve treatment outcomes. Through predictive analytics, we can pinpoint individuals at the highest risk, foresee potential complications, and optimize care. The doctors can determine each patient's most effective treatment course, leading to improved outcomes. Consequently, a computer-aided diagnosis (CAD) system can aid physicians in decision-making for early diabetes detection. The CAD system assesses blood sugar levels, hemoglobin A1c levels, and other relevant clinical data to help doctors diagnose diabetes and recommend appropriate actions based on the gathered information.

Among the various applications of Artificial Intelligence, the healthcare industry has witnessed some of the most promising outcomes from the evolving field of machine learning. Numerous strategies, including data mining and machine learning, have been presented in the literature to predict the early onset of diabetes. Some notable works include that of *Haritha, Babu & Sammulal (2018)*, which proposed a solution for diabetes detection using Cuckoo Search (CS) techniques with K nearest neigbour (KNN). *Benbelkacem & Atmani (2019)* developed SVM, naïve Bayes (NB), and RF models for diabetes detection. *Marzouk, Alluhaidan & El Rahman (2022)* selected optimal features in PIMA using Autoencoder and applied deep neural network (DNN). The PIMA dataset is frequently used in machine learning and predictive analytics to develop and assess diabetes diagnosis systems. This dataset contains medical records from a study conducted on 768 women in Pima, India, in the 1990s by the National Institute of Diabetes and Digestive and Kidney Diseases. Similarly, *Jaganathan et al. (2022)* employed a DNN for early diabetes detection. Key stages, including feature selection and data standardization, are involved in developing an accurate predictive model for diabetes detection. While feature selection helps decide which aspects are most important to include in the model, data normalization ensures that all features are scaled uniformly. These steps can help enhance the model's accuracy and stability, increasing its utility for forecasting diabetes. The proposed work standardizes the PIMA and early-risk diabetes datasets using the standard scalar standardization technique. To the best of our knowledge, we are the first to employ the Sequential Feature Selection method for diabetes prediction in the literature. Features are selected based on their effectiveness in improving the model's performance using the SFS subset selection technique.

This study employs a machine learning technique on the PIMA and early diabetes risk datasets. In the initial phase, optimal features were selected from both datasets using Sequential Feature Selection and other techniques. The data were then normalized using the standard scalar. Subsequently, a classification model was developed for early diabetes detection based on RF, GB, ANN, Tab-net, and SVM on the PIMA and early-risk diabetes datasets. This study aims to enhance the prediction performance in diabetes classification

by selecting relevant features using SFS and other methods and by searching for the best hyperparameters for the machine learning model through hyperparameter tuning. We hypothesize that this proposed method will improve the performance of the PIMA and early-risk diabetes datasets. The primary goal is to enhance the performance of diabetes tasks on both datasets. Our research introduces a cutting-edge, machine learning-based strategy for anticipating diabetes with significant clinical utility and promising practical application. This approach addresses several significant challenges in diabetic treatment, improving patient outcomes and more efficient healthcare delivery using advanced computational methods.

The clinical relevance and real-world applicability are listed below:

- **Personalized treatment strategies:** Diabetes manifests differently in various individuals, making it a complex and heterogeneous condition. This technology enables the creation of customized treatment programs based on medical history for specific patients. This individualized strategy has the potential to significantly improve glycemic control, reduce the risk of complications, and enhance the overall quality of life for patients.
- **Early identification and intervention:** One of the most essential aspects of controlling diabetes is preventing complications through early detection and action. Our machine-learning model was developed using a large clinical dataset and continuous glucose monitors. This allows for the early detection of trends and patterns indicating poor glycemic control. Our technology alerts individuals early, enabling healthcare professionals to intervene swiftly and prevent negative outcomes.
- **Resource allocation and efficiency:** Diabetes has a significant financial impact on healthcare systems. By identifying patients more likely to experience difficulties or unsuccessful outcomes, our method helps optimize resource allocation. Healthcare practitioners can efficiently allocate resources by risk-stratifying their patients, ensuring interventions are prioritized for those who would benefit most. This promotes patient care and the efficiency of healthcare delivery.

The key contributions of this study are listed below:

- This study conducted experiments using tree-based methods (RF, GB) and deep learning methods (TabNet and ANN) on the PIMA and Early Stage Diabetes datasets.
- The study performed hyperparameter optimization using grid search.
- The study proposed the SFS method with a backward strategy to select relevant features.

The rest of the article's organization is as follows: "Literature Review" provides an overview of related work on diabetes prediction. "Proposed Methodology for Diabetes Prediction" presents the complete methodology of our proposed work. "Results and Discussion" discusses and interprets our results, and we conclude our work in "Conclusions and Feature Scope".

## LITERATURE REVIEW

This section reviews previous efforts to use machine learning for early-stage diabetes prediction.

### Diabetes prediction on PIMA data set

A predictive model based on machine learning and deep learning utilizes risk factors as input features to predict whether patients have diabetes. This computer-aided system assists physicians in decision-making. *Haritha, Babu & Sammulal (2018)* employed a hybrid approach, predicting diabetes patients using Firefly and cuckoo search techniques for feature selection. Among these techniques, Cuckoo-Fuzzy-KNN achieved the highest accuracy. *Maniruzzaman et al. (2018)* compared six feature selection techniques (principal component analysis, analysis of variance, Fisher discriminant ratio, random forest, logistic regression, mutual information) and ten machine learning classifiers on a dataset of 768 diabetes patients. *Benbelkacem & Atmani (2019)* used an RF classifier to predict diabetes patients and compared its performance with other classifiers, finding a lower error rate than SVM and C4.5. *Reddy et al. (2020)* employed NB, RF, SVM, GB, KNN, and logistic regression (LR) to classify diabetes in PIMA Indian patients.

To extract the most useful features from the diabetes data set, the research authors (*Arora et al., 2022*) employed the K-means clustering technique. To classify people with diabetes, they used an SVM Classifier on the collected features. According to the findings of this study, this method has a 98.7% accuracy.

Using the patient's QR card, researchers in the study (*Marzouk, Alluhaidan & El Rahman, 2022*) developed a system for monitoring diabetes and communicating treatment updates to medical professionals. The author performed an experiment on PIMA and diabetes synthetic data sets. RF, NB, LR, and ANN are all utilized in the diabetes categorization. The experimental result shows that the ANN achieved 81.6% accuracy. In a article (*Patil & Ingle, 2021*), KNN, NB, SVM, and RF classifiers for the classification of diabetes are compared, and the results are compared with ANN. Their proposed method achieved 97.66% accuracy for the classification of diabetes. A study (*Jaganathan et al., 2022*) used KNN, RF, NB, and LR to classify diabetes. RF achieved 89.58% accuracy, which is the highest compared to another method. In a study, the author (*Chang et al., 2023*) used a DT, NB, and an RF classifier to classify diabetes. The prediction accuracy of random forest is 85.17%.

### Diabetes prediction with neural network

To acquire precise prediction results for a real-world problem, neural networks can be used in many contexts. Its versatility means it may be used in various contexts where it will perform admirably and yield the desired results for solving practical problems. For instance, *Pradhan et al. (2020)* used a Neural Network to classify the patients as sick or healthy 768 diabetes patients. The study employed a standard backpropagation method for ANN model training. The experimental results verified that ANN classifies patients with an 87% accuracy rate. Optimal feature selection plays a vital role in predictive modeling. *Kannadasan, Edla & Kuppili (2019)* selected optimal features using a stacked autoencoder

from the PIMA 768 data set and applied DNN for prediction. The model is evaluated using precision, recall, specificity, and F1 measures. The experimental results showed that this model achieved 86.26% accuracy, 87.92% recall, and 90.66% precision. *Naz & Ahuja (2020)* applied deep neural networks (DNN), NB, and DT. The experimental results show that deep learning achieved the highest accuracy compared to another classifier used in this proposed work with 98% accuracy. *Ayon & Islam (2019)* also applied a DNN from the Indian PIMA data set to detect diabetes. The model's performance is evaluated using two measurement parameters, F1 and Accuracy.

The blood vessels in the retina can be damaged due to diabetic retinopathy. *Alfian et al. (2020)* utilized a DNN to predict diabetic retinopathy in patients, employing recursive feature selection and elimination techniques to eliminate irrelevant attributes. The experimental results demonstrated that the DNN predicted retinopathy with an accuracy of 82.03%. Similarly, *Gadekallu et al. (2020)* predicted retinopathy from risk factors using DNN, extracting significant features with principal component analysis and normalizing raw data with a standard scalar. The authors also reduced the dimension of data using the Firefly algorithm. Early detection of end-stage renal disease can benefit patients with kidney disease, protecting them from developing adverse health conditions. In clinical trials, *Belur Nagaraj et al. (2020)* utilized LR, SVM, RF, feed-forward neural network (FFNN), Cox proportional hazard regression, and the KFRE model. The proposed FFNN approach outperformed the baseline scheme. *Kowsher et al. (2019)* compared ANN, RF, NB, and DT classifiers using information from 9,483 diabetic patients. In another study (*Tan et al., 2022*), the authors proposed a genetic algorithm (GA) based on a stacking technique for diabetes prediction. They selected relevant features using a GA, applied neural network and SVM as baseline learners for meta-models, and conducted experiments on the Early-Stage Diabetes Risk Prediction dataset. Their proposed method achieved a remarkable 98.71% accuracy on the Early-Stage Diabetes Risk Prediction dataset.

## Ensemble learning and decision tree for diabetes prediction

*Abedini, Bijari & Banirostam (2020)* applied an ensemble hierarchical model for diabetic patient classification. Firstly, LR and DT are trained, and the output is fed to a neural network. This approach classifies the diabetes patients with 82% accuracy from the PIMA diabetes data set. *Pei et al. (2019)* identify the potential of Type II diabetes using a DT classifier to identify diabetes. The experimental results show that the proposed model achieved 94.8% AUC, 94.2% accuracy, 94.0% precision, and 94.2% recall value. *Maniruzzaman et al. (2020)* identified the risk factors for diabetes disease based on *p*-value and odd ratios. Adaboost, NB, RF, and DT were applied to predict the diabetes patients. The experimental results show that LR and RF combination perform better.

*Sneha & Gangil (2019)* used optimal features for early diabetes detection and applied RF, SVM, NB, DT, and KNN on 2,500 data items. The experimental results show that SVM achieved the highest accuracy, which is 77.73%. *Nibareke & Laassiri (2020)* compared DT, LR, and NB classifiers on Vincent Sigillitan's data set. This work aims to overview machine

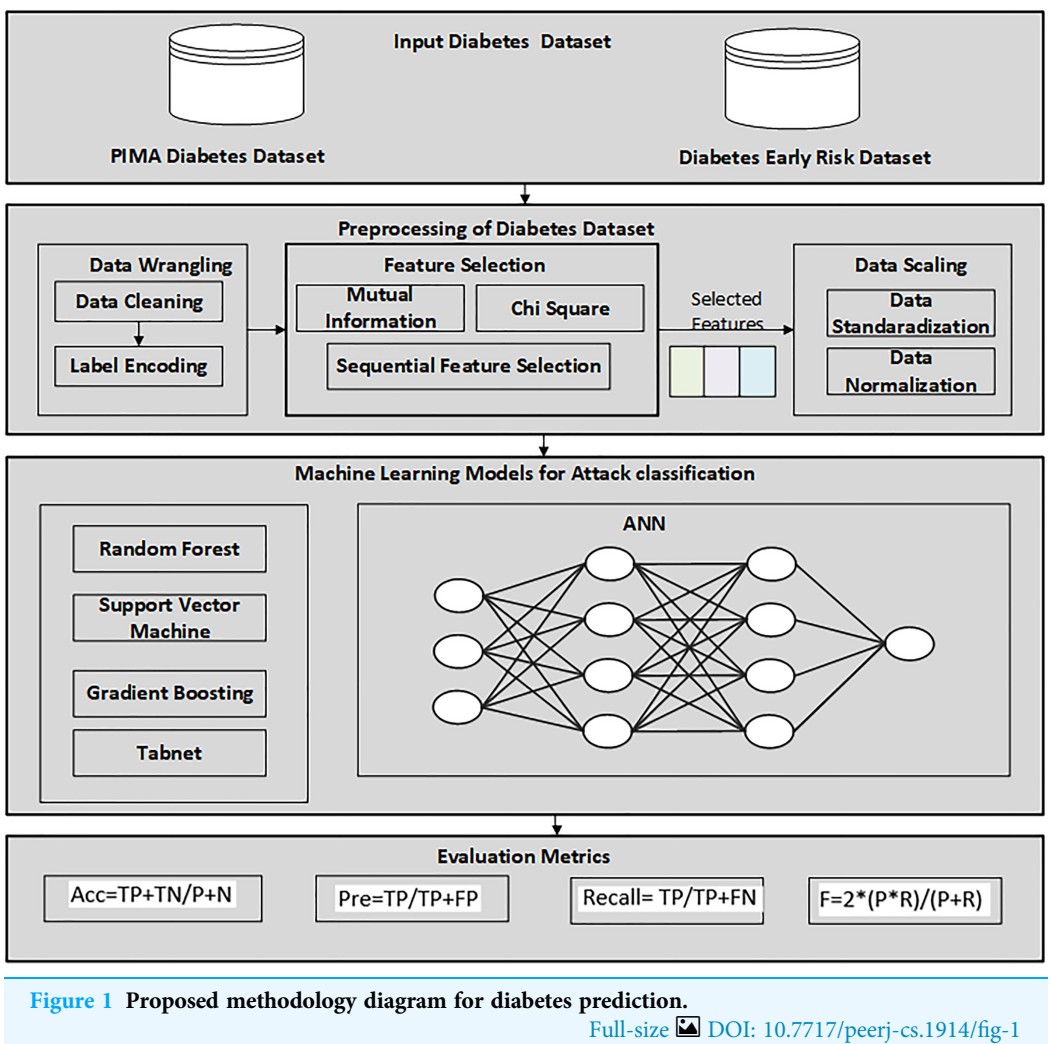

**Figure 1 Proposed methodology diagram for diabetes prediction.**

learning modeling and big data tools. The experimental results show that the decision tree achieved 0.766% accuracy and 0.125 RMS value.

# PROPOSED METHODOLOGY FOR DIABETES PREDICTION

Firstly, the diabetes data set has been collected from UCI, the machine learning repository. After this, a data preprocessing technique is applied to prepare data and select the relevant features for the machine learning classifier. The reason for using the feature selection method is to select the relevant features that help to enhance the performance. RF, GB, ANN, TabNet, and SVM were then utilized to detect diabetes patients from the dataset. Finally, the proposed SFS-based framework was evaluated using various performance evaluation measures and compared with other feature selection methods. Figure 1 illustrates the methodology diagram for the proposed work in diabetes prediction. A 10-fold cross-validation was employed to validate the model and mitigate overfitting. This method partitions the data into 10 equal-size folds, using each fold once as validation while the remaining k-1 folds are utilized for training.

## Datasets

This study used two datasets for diabetes prediction.

### PIMA Indian dataset

The PIMA Indian dataset comprises specific parameters crucial for the early prediction of diabetes. With 768 instances, it includes eight influential attributes contributing to diabetes prediction. The target attribute is binary, encompassing 268 diabetes samples and 500 non-diabetes samples. Refer to Table 1 for a detailed dataset description.

### Early-stage diabetes risk prediction dataset

The Early-Stage Diabetes Risk dataset comprises sixteen essential features for diabetes prediction, with 520 instances and 16 input features. The target attribute is binary. Refer to Table 2 for a detailed description of each attribute. To prepare both datasets, we underwent specific pre-processing steps, detailed in the next section.

---

**Algorithm 1:** A Robust Diabetes Classification Framework Based on Deep Learning and Ensemble Learning Method

---

**Data:** Diabetes Dataset $diabF = (f_1, f_2, f_3, ... f_n)$, feature dimension=n, target feature size=k
**Result:** Diabetic, Normal
initialization;
$diabF \leftarrow (READCSV)$;
$Feature \leftarrow SplitFeature(diabF)$
**for** *each Feature* **do**
    **if** *AlphaNumericFeatures* **then**
        $diabF \leftarrow FeatureEncoding(diabF)$
    $diabF \leftarrow standardScalar(data)$
$T_{suboptimal} \leftarrow diabF$
**for** *i from 1 to n-k* **do**
    $f = min(J(T_{suboptimal} - \{f_i\}))$
    $T_{suboptimal} \leftarrow T_{suboptimal} - \{f\}$
$trainset, testset \leftarrow Datasplit(T_{suboptimal})$
$Models \leftarrow InitilizeParameters()$
**for** *each* **Models** **do**
    $Model \leftarrow ModelTraining(trainset)$;
    $\hat{Y} \leftarrow ModelPrediction(testset)$;
    Evaluate;
    $Accuracy \leftarrow TP + TN/(TP + FN + FP + TN)$
    $Precision \leftarrow TP/(TP + FP)$
    $Recall \leftarrow TP/(TP + FN)$
    $FScore \leftarrow 2*(PR)/(P + R)$

---

## Data preprocessing

Data preprocessing is the procedure of cleaning and transforming raw data. This work uses label encoding to convert categorical labels into discrete forms. Additionally, we normalize the PIMA dataset using the standard scaler normalization technique (*Sharma et al., 2021*). The equation for the standard scaler is provided below:

$$Z = Xi - \mu/\sigma \tag{1}$$

$\mu$ is a mean of a given data and $\sigma$ is variance of a given data. While X_i is an input feature.

**Table 1 Brief description of PIMA's features.**

| F. No | Feature name | Description | Values |
|---|---|---|---|
| F1 | Glucose | Glucose is a monosaccharide | 0–199 |
| F2 | Skin thickness | Skin fold thickness | 0–99 |
| F3 | BMI | Body mass index | 0–67.1 |
| F4 | Diabetes pedigree function | Synthesis of the diabetes history in relatives | 0.078–2.42 |
| F5 | Pregnancy | Total number of time Participant pregnant | 0–17 |
| F6 | Serum insulin | serum insulin | 0–846 |
| F7 | Age | Age of participant | 21–81 |
| F8 | Diastolic blood pressure | Diastolic blood pressure | 0–122 |
| F9 | Outcome | Target class attribute | Yes or No |

**Table 2 Brief description of early diabetes detection dataset's features.**

| F. No | Feature name | Description | Values |
|---|---|---|---|
| F1 | Age | Age of an patients | 20–65 |
| F2 | Sex | Gender of an patient | Male, female |
| F3 | Polydipsia | excess drinking | Yes, No |
| F4 | Sudden weight loss | Describe is the patient is losing weights or not | Yes, No |
| F5 | Weakness | Patient weakness | Yes, No |
| F6 | Polyphagia | extreme hunger | Yes, No |
| F7 | Genital thrush | Affected by a type of yeast called Candida. | Yes, No |
| F8 | Visual blurring | Limited type of blindness | Yes, No |
| F9 | Itching | Itching disease cause by a toxin | Yes, No |
| F10 | Irritability | Feeling of agitation | Yes, No |
| F11 | delayed healing | Wound healing | Yes, No |
| F12 | partial paresis | Partial weakening of muscles | Yes, No |
| F13 | Muscle stiness | Muscle, pains, and cramping | Yes, No |
| F14 | Alopecia | Hair falls in small patches | Yes, No |
| F15 | Obesity | Excessive amount of body pain | Yes, No |
| F16 | Polyuria | Large amounts of urine | Yes, No |
| F17 | Target attribute | Target attribute | Yes, No |

### Feature selection using sequential feature selector

Sequential feature selection is a method used in feature engineering to select **K** features from the **S** feature set. It is a greedy feature selection procedure that reduces the n-dimensional feature space to a k-dimensional feature space, where **K** is less than **d** (k ¡ S). The method iteratively evaluates different feature subsets and selects features based on their performance. In each iteration of SFS, the best new feature is chosen based on cross-validation scores and added to the selected feature list. SFS starts with an empty set and adds or removes one feature at a time until a feature subset reaches the desired k. This work uses a backward sequential feature selection (BSFS) technique to select the most relevant

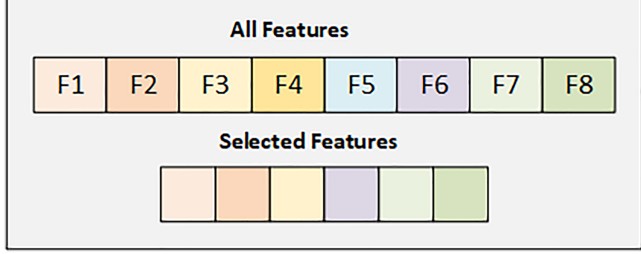

**Figure 2** Graphical representation of selected feature from PIMA dataset.

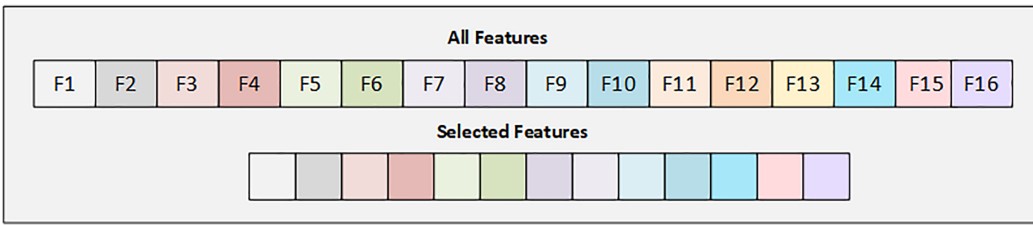

**Figure 3** Graphical representation of selected feature from early diabetes risk dataset.

features. BSFS begins with the entire feature set and gradually removes one feature at a time. The model is trained on the remaining features, and the performance is validated. Features with the smallest impact on prediction performance are eliminated until the desired **k** features are selected. This method enhances efficiency, reduces dimensionality, and focuses on more informative features that contribute positively to diabetes prediction.

## Feature selection strategy on Pima dataset

In the PIMA dataset, we have eight features, and we select the best six features using the SFS backward strategy. Figure 2 details the selected and dropped features.

Figure 2 displays details of all features and the selected features from the PIMA dataset. Selected attributes are highlighted with their respective colors.

## Feature selection strategy on early risk diabetes dataset

In the Early Risk Diabetes dataset, we have 16 features, and we selected the best 12 features using the SFS backward strategy. Figure 3 illustrates the details of the selected and dropped features.

Figure 3 displays the selected features from the Early Diabetes Risk dataset, with each attribute highlighted in its respective color.

### Chi-square

The Chi-square feature selection approach (*Mushtaq et al., 2020*) is employed in this study to assess the association between diabetes features and the target outcome class. This statistical test evaluates the degree of independence between two categorical variables. It helps determine whether there is a significant correlation between each feature and the

target variable during feature selection. Features strongly correlated with the target variable are retained, as they are essential for classification, while features with weaker associations may be eliminated.

### Mutual information

Mutual information (MI) feature selection (*Dai & Chen, 2020*) is a powerful method for uncovering hidden correlations between variables. MI estimates the information shared between a feature and a target variable, proving highly beneficial in machine learning applications. MI can identify complex relationships, including non-linear and interactive linkages, by measuring the reduction in uncertainty of one variable when the value of another is known. The relevance of each feature to the target variable is evaluated during the selection of MI features based on the information they share. After selecting features using various methods, we feed the data into a machine-learning model. Details of the model are presented in the next section.

## Machine learning method for diabetes prediction

ANN, RF, Tab-net, SVM, and GB are employed for diabetes prediction, and the parameters of each method are tuned using grid search. Grid search is a hyperparameter tuning method that optimizes the hyperparameters of each method. In grid search, we first define a range of hyperparameters, and the machine learning model is evaluated for possible combinations. The hyperparameter yielding the best performance on the validation data is then selected for the final training of the model.

### Artificial neural network

We employed an ANN for diabetes prediction. In recent years, ANNs have proven highly successful in solving classification problems. An ANN is a fully connected neural network where each unit is connected to the previous unit, receiving input from the previous layer. Each unit has its weight and bias. The input layer takes input data, and the output layer produces the final output. The intervening layers are referred to as hidden layers. The net input is calculated by multiplying the weight with the respective input, and each unit in the hidden layer applies an activation function to the input (*Li et al., 2021*). The equation for net input and the activation function is described in Eq. (2).

$$a = actfunction(wx + b). \tag{2}$$

W is a weight matrix, and x is an input variable. The **actfunction** represents the activation function. Our work utilizes the Rectified Linear Unit (ReLU) activation function, a nonlinear activation function (*Pesch et al., 2022*). The standard backpropagation trains the neural network by comparing the target value with the predicted result and updating the weight of the training pair to minimize the loss function (*Rosenbaum, 2022*). Since diabetes prediction is a binary classification problem, the loss function is described in Eq. (3).

$$Error = (Ypred - Yactual)^2. \tag{3}$$

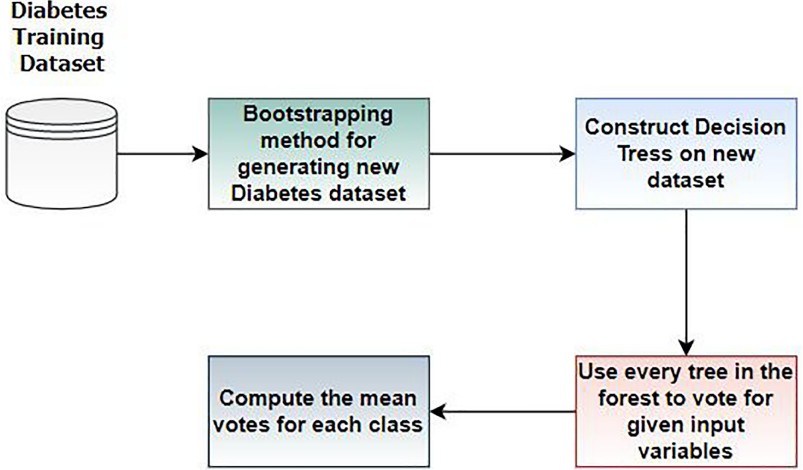

**Figure 4 Steps of random forest for diabetes prediction.**

Ypred represents the predicted value, and yactual shows the actual value. Backpropagation is performed using the Adam optimizer (*Bijukchhe et al., 2022*), which utilizes momentum and RMSprop to learn new weights, as presented in Eq. (4).

$$Wnew = Wold - \alpha Vdw^{corr} \qquad (4)$$

Wnew represents a newly updated weight, and Wold describes an old weight. $Vdw^{corr}$ is the derivative of the gradient using RMS and momentum, where $\alpha$ is the learning rate. The main reason behind using this method is to handle complex problems when dealing with nonlinear surfaces.

### Random forest

RF is an ensemble learning classifier consisting of multiple decision trees, proposed by Breiman and Adele Cutler (*Khan et al., 2022*). RF is widely used in various application areas, including medical diagnosis (*Solaiyappan & Wen, 2022*). In this work, we employed RF for diabetes prediction. Figure 4 illustrates the workflow steps of random forest.

In Step 1, we divide the dataset into training and testing sets. The bootstrapping method creates new datasets from the training set. In Step 2, a decision tree is constructed based on the results of Step 1. In Step 3, multiple DT are built to form a random forest, and Steps 1 and 2 are repeated. In this step, each tree in the forest votes for the given variables, and the mean vote for each class is computed. RF has various hyperparameters requiring optimal tuning of results. We train RF for diabetes prediction using these parameters, as discussed in Table 3. The rationale for choosing this method is based on the assumption that Random Forest performs well with tabular data, such as the PIMA and early-risk diabetes datasets.

### Gradient boosting

GB is an ensemble learning classifier that combines numerous weak learning models to create a robust predictive model (*Malik et al., 2022*). A DT is typically used as a weak

**Table 3 Hyperparameter of random forest for diabetes prediction.**

| Hyper parameter | Values |
|---|---|
| Random state | None |
| Min samples leaf | 1 |
| Min samples split | 2 |
| Max features | Auto |
| Bootstrap | True |
| N estimators | 50 |
| Criterion | Gini |

**Table 4 Hyperparameter of gradient boosting for diabetes prediction.**

| Hyper parameter | Values |
|---|---|
| Min samples leaf | 1 |
| Max depth | 4 |
| n estimators | 19 |
| Min samples split | 2 |
| Max features | 2 |
| Learning rate | 0.5 |
| Random state | None |

learner in GB and performs well on tabular data. The GB algorithm comprises three main elements: a loss function, a weak learner, and an additive model to incorporate a weak learner. It applies to both classification and regression problems. Gradient boosting involves various hyperparameters that require tuning for optimal results. In our study, we utilized gradient boosting for diabetes prediction, and the hyperparameters tuned for diabetes prediction are discussed in Table 4.

### TabNet

This study employed the TabNet model (*Khan et al., 2023*) for diabetes classification, a deep learning model designed explicitly for tabular data. The utilization of this method capitalizes on its ability for diabetes prediction. The approach is distinguished by its sequential attention to specific aspects. Notably, it employs instance-wise feature selection, allowing each row in the training dataset to have different features. The TabNet model incorporates soft feature selection due to its integrated deep learning architecture. A notable quality of TabNet is its capability to provide both local and global interpretability, which is crucial for understanding its predictions. The model is trained using specific parameters explicitly tailored for diabetes.

### Support vector machine

A SVM (*Bansal, Goyal & Choudhary, 2022*) is employed for diabetes classification, constructing a hyperplane in a multi-dimensional space to separate classes (diabetic and non-diabetic). By iteratively creating an optimal hyperplane, it reduces inaccuracy. The

SVM model is fed with the PIMA and Early Risk Diabetes datasets, utilizing the kernelling method to transform the low-dimensional space into a higher-dimensional one. Given the nonlinear nature of the diabetes dataset, SVM serves as an effective classifier, particularly for solving nonlinear problems through the kernel approach. The gamma parameter is set to 0.7, employing the RBF kernel. Nonlinear kernels, such as RBF, perform better for nonlinear problems. Evaluating the performance of the machine learning model is crucial for validating its effectiveness. Details of the evaluation metrics are provided in the next section.

## Evaluation metrics for diabetes prediction

When training classifiers to address practical problems, they learn from the given environment and make predictions. Classifier performance evaluation is essential to determine the accuracy of class label predictions. Various evaluation metrics are utilized in diabetes prediction, including accuracy, precision, recall, and F-measure.

### *Accuracy*

The correct prediction rate in classification problems, known as accuracy, is calculated by dividing the accurate classifier predictions by the total number of predictions.

$$Accuracy = Correct\ Prediction/Total. \tag{5}$$

### *Precision*

Precision is the ratio of correct positive predictions to the total predicted positive instances. When there are no false positives and only true positives, the precision of the model is equal to one.

$$Precision = TP/(TP + FP). \tag{6}$$

### *Recall*

Recall is the ratio of correct positive predictions to the sum of false and correct positives. It indicates how accurately the model predicts actual positives.

$$Recall = TP/(TP + FN). \tag{7}$$

### *F measure*

F-measure is the harmonic mean of recall and precision. While recall and precision provide valuable insights, F-measure captures the combined properties, offering a more comprehensive evaluation of predictions.

$$F\ Mesaure = 2*(PR)/(P + R). \tag{8}$$

## RESULTS AND DISCUSSION

This study proposes a machine learning method for classifying diabetes and conducts experiments on two publicly available datasets, the PIMA and Early-Risk diabetes datasets.

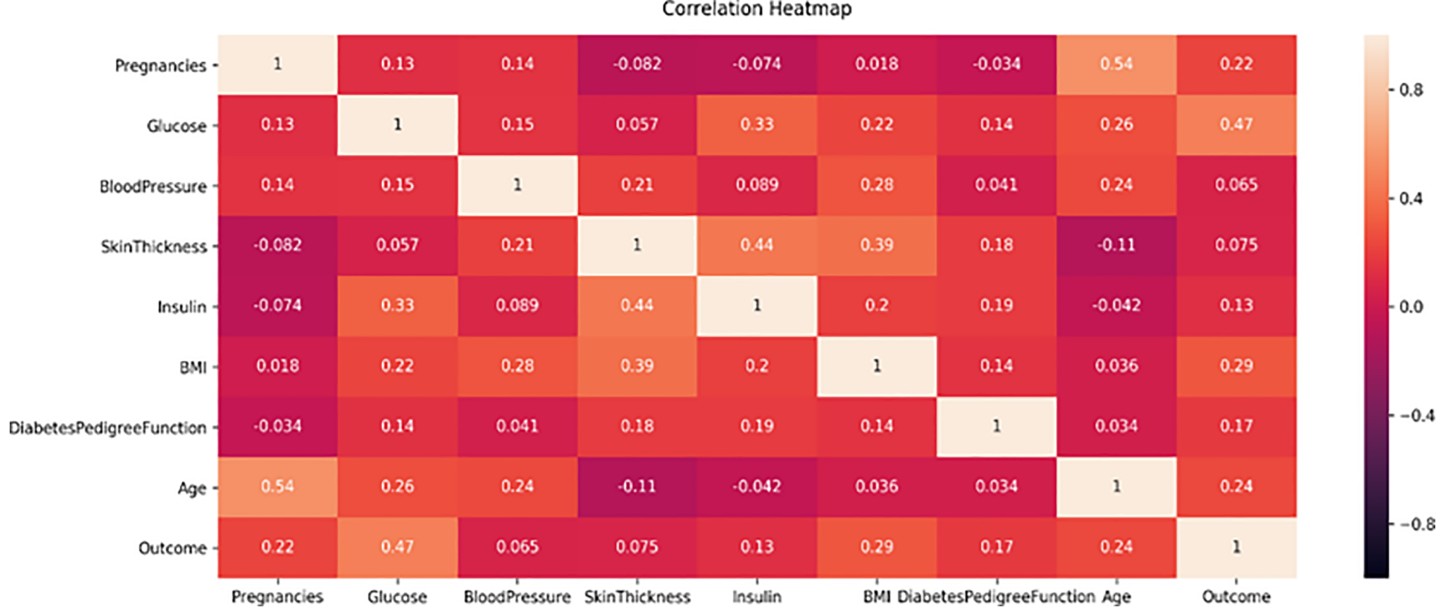

**Figure 5** PIMA dataset feature correlation heatmap.               

In the initial phase, the study analyzes the correlation of both datasets. Correlation analysis, a statistical method, examines the relationship between independent input features and the target class, which can be positive or negative. A correlation heat map, a graphical representation of correlation, illustrates the strength of the relationship between each independent variable and the target outcome class label. Figure 5 displays the correlation analysis of the PIMA dataset, while Fig. 6 shows the correlation analysis of the Early Risk Diabetes dataset. Subsequently, the study employs several feature selection methods to choose relevant features, including chi-square, MI, and SFS. The selected features are input into RF, ANN, GB, SVM, and Tab-net models to predict diabetes. Experimental results, with and without feature selection, are compared to assess the impact of feature selection methods on diabetes classification and evaluate the performance against existing research. This work demonstrates improved diabetes prediction using ANN and RF. The experimental results for both datasets, with and without feature selection, are presented in the following subsections. Details of the experimental setup are provided in Table 5.

## Correlation analysis

Figure 5 displays the correlation heatmap of the PIMA diabetes dataset. It indicates that the blood pressure feature has a low correlation with the outcome variable, while another feature exhibits a high correlation with the target class label.

Figure 6 displays the correlation heatmap of features in the early-stage diabetes risk dataset. The gender feature shows a low correlation with the target attribute, while another feature exhibits a high correlation with the target class label.

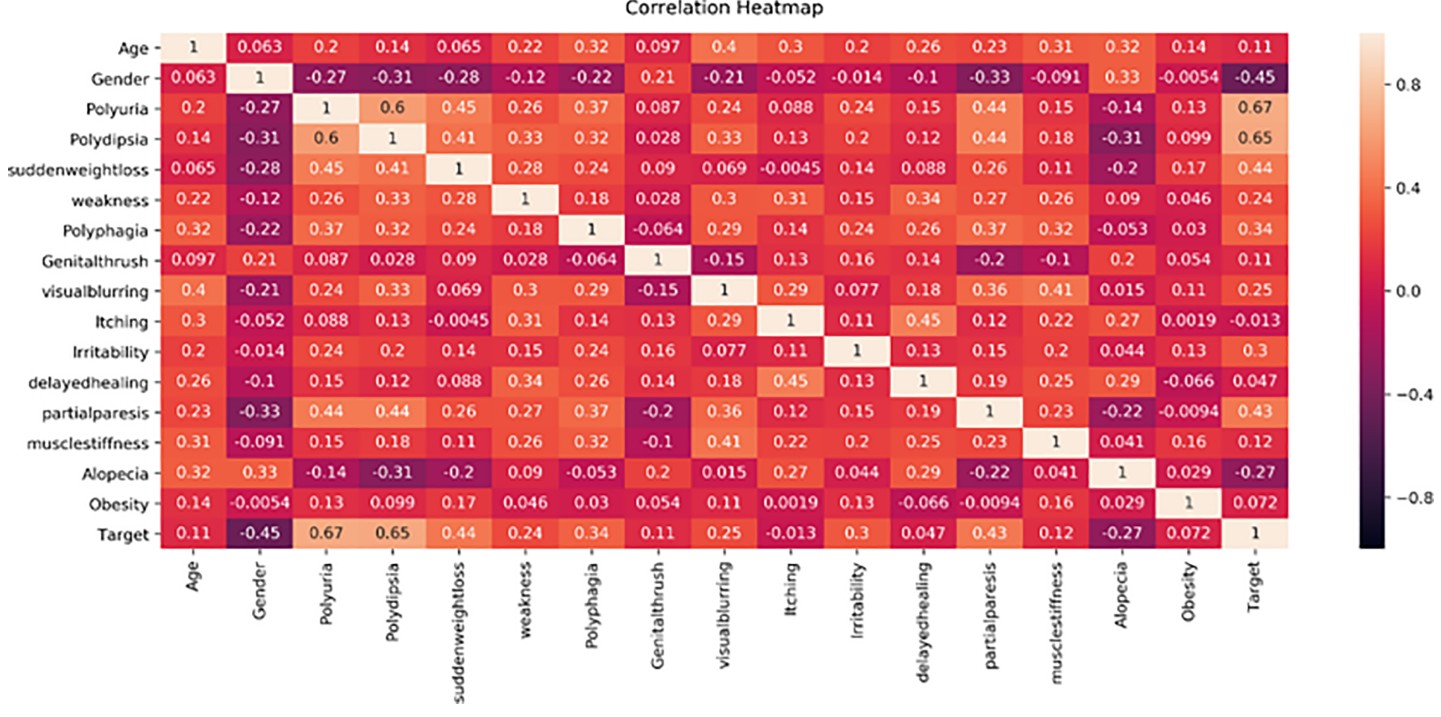

**Figure 6 Early diabetes dataset feature's correlation heath map.**           

| S. No. | Components | Detail |
|---|---|---|
| | Table 5 Experimental setup detail. | |
| 1 | Hardware | Intel Core i7 7th Gen PC |
| 2 | Operating system | Window 11 |
| 3 | Primary storage | 8 GB RAM |
| 4 | Data file storage | MS Excel |
| 5 | Programming language | Python |
| 6 | Python required libraries | Pandas, Scikit, Tensorflow, Seaborn, Matplotlib |
| 7 | IDE | Jupyter Notebook |

## Experimental results without feature selection

Table 6 presents the results of machine learning techniques for diabetes prediction on the PIMA and early diabetes risk datasets using all features. The ANN achieved an accuracy of 98.71%, and the RF achieved 96.79% accuracy on both the PIMA and Early-Risk Diabetes datasets with all available features. It suggests that ANN and RF outperform other models in predicting both datasets. However, the TabNet model exhibits lower performance on the early-risk diabetes dataset than other methods while showing better prediction results than RF, GB, and SVM on the PIMA dataset. These variations in model performance are attributed to their learning capacities, with ANN and RF demonstrating better capabilities in learning complex decision boundaries for intricate non-linear problems.

**Table 6 Experimental results of machine learning model without feature selection.**

| Method | Accuracy | Precision | Recall | F score |
|--------|----------|-----------|--------|---------|
| Experimental results of the PIMA dataset | | | | |
| RF | 70.13% | 70.18% | 70.12% | 69.12% |
| GB | 70.56% | 70.45% | 70.6% | 69.83% |
| SVM | 71.43% | 73.14% | 71.42% | 69.5% |
| TabNet | 82.683% | 82.947% | 82.693% | 81.914% |
| ANN | 98.71% | 98.74% | 98.84% | 98.79% |
| Experimental results of the early diabetes risk dataset | | | | |
| RF | 96.79% | 96.84% | 96.81% | 96.78% |
| GB | 94.23% | 94.22% | 94.24% | 94.2% |
| SVM | 92.95% | 93.08% | 92.95% | 92.98% |
| TabNet | 84.615% | 84.935% | 84.615% | 84.213% |
| ANN | 96.41% | 96.25% | 96.31% | 96.27% |

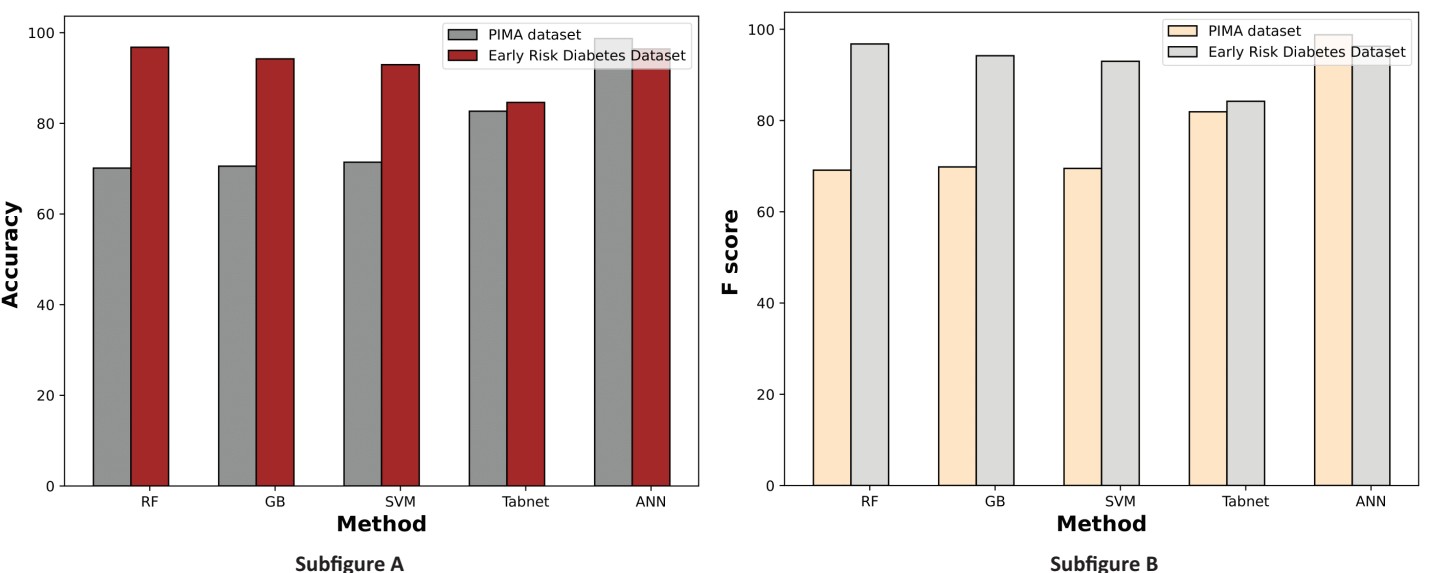

Subfigure A

Subfigure B

**Figure 7 Comparison of machine learning methods based on an accuracy and F score (without feature selection).**

Figure 7 displays a comparative graph of results for machine learning methods on the PIMA and Early Risk diabetes datasets without feature selection. The graph contrasts the accuracy and F score of different machine learning methods, revealing that the ANN outperformed other methods on the PIMA dataset. At the same time, RF excelled on the Early Risk Diabetes dataset.

The confusion matrices of the machine learning model for both datasets are presented in Fig. 8 without feature selection. It indicates that RF has a low misclassification rate on

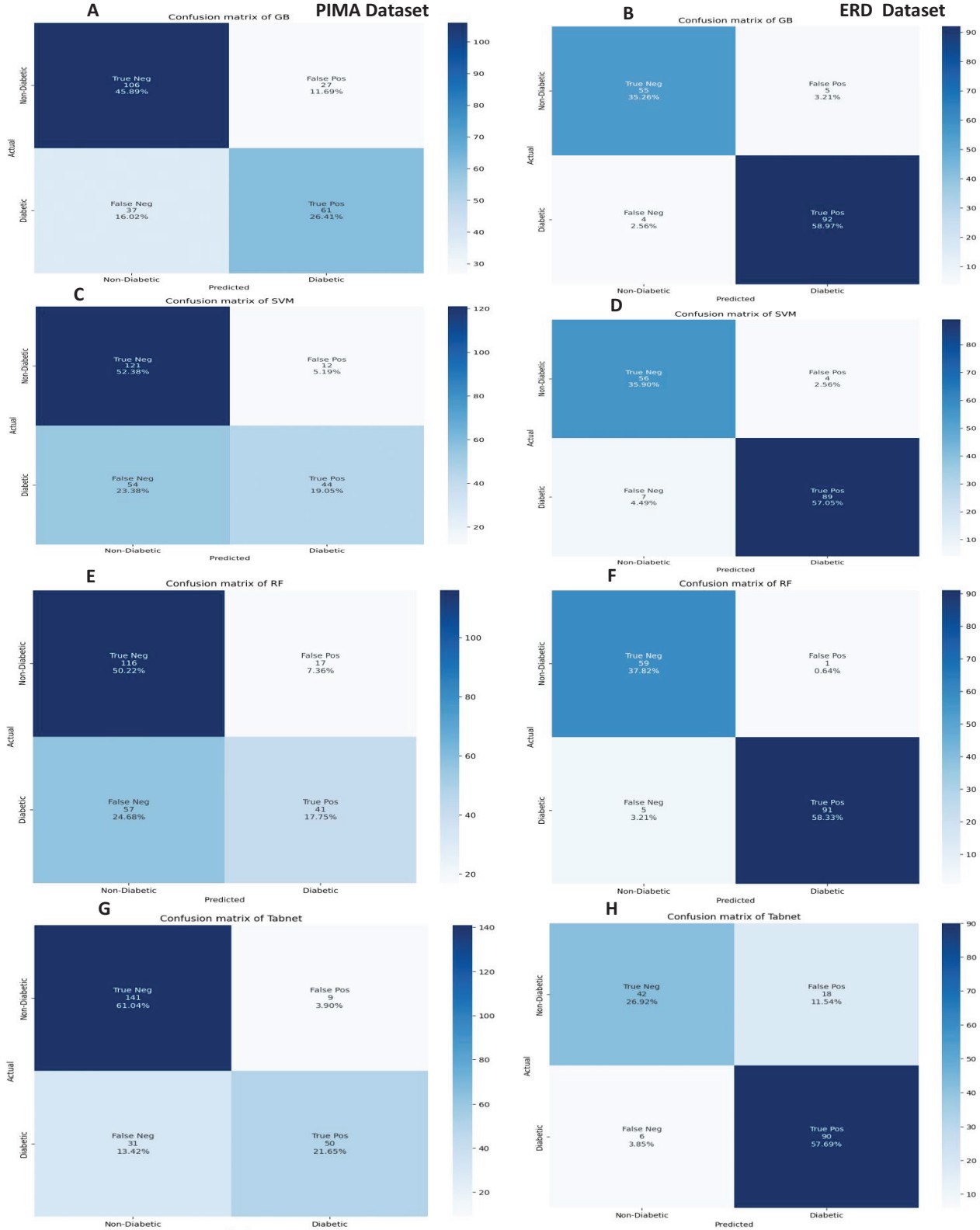

**Figure 8 Confusion metrics of machine learning models (without feature selection).**

**Table 7 Experimental results of chi-square feature selection.**

| Method | Accuracy | Precision | Recall | F score |
|---|---|---|---|---|
| Experimental results of the PIMA dataset | | | | |
| RF | 68.4% | 68.14% | 68.41% | 67.56% |
| GB | 68.83% | 68.55% | 68.84% | 68.15% |
| SVM | 70.56% | 70.84% | 70.55% | 69.39% |
| TabNet | 84.848% | 88.189% | 84.84% | 85.18% |
| ANN | 98.69% | 98.72% | 98.79% | 98.8% |
| Experimental results of the early diabetes risk dataset | | | | |
| RF | 98.07% | 98.08% | 98.1% | 98.07% |
| GB | 98.72% | 98.74% | 98.72% | 98.71% |
| SVM | 93.59% | 93.59% | 93.59% | 93.59% |
| TabNet | 86.538% | 88.954% | 86.538% | 85.774% |
| ANN | 94.87% | 94.90% | 96.88% | 96% |

the Early Risk Diabetes dataset. However, the Tab-net model exhibits a lower misclassification rate on the PIMA dataset than other machine learning models.

## Experimental results of chi-square feature selection

The experimental results of the chi-square feature selection are presented in Table 7. It illustrates that the prediction accuracy of ANN and RF is 98.69% and 98.07%, respectively, on the PIMA and Early Diabetes Risk datasets. The performance of both models has improved compared to the performance without feature selection. On the other hand, the performance of some models has decreased on the selected features of the chi-square feature selection. The reason for the reduction in the performance of some methods is that the learning capacity of ML models varies due to input attributes and their relationship with the target class label. The overall prediction performance of ANN and RF with the chi-square feature selection is noteworthy.

Figure 9 demonstrates the machine learning results using the chi-square feature selection. It indicates that the prediction performance of ANN and RF is high on both datasets, respectively. With this feature selection method, most machine learning models show improved prediction results on both datasets.

## Experimental results of MI feature selection

The experimental results of MI feature selection are described in Table 8. It demonstrates that the machine learning model's performance is improved on both datasets with this feature selection method compared to the chi-square feature selection. The prediction accuracy of ANN and RF is also enhanced in both datasets.

The experimental results of machine learning models with MI feature selection are presented in Fig. 10. This figure illustrates that the machine learning model's performance is improved in terms of accuracy and F-score with the MI feature selection method compared to chi-square and without feature selection.

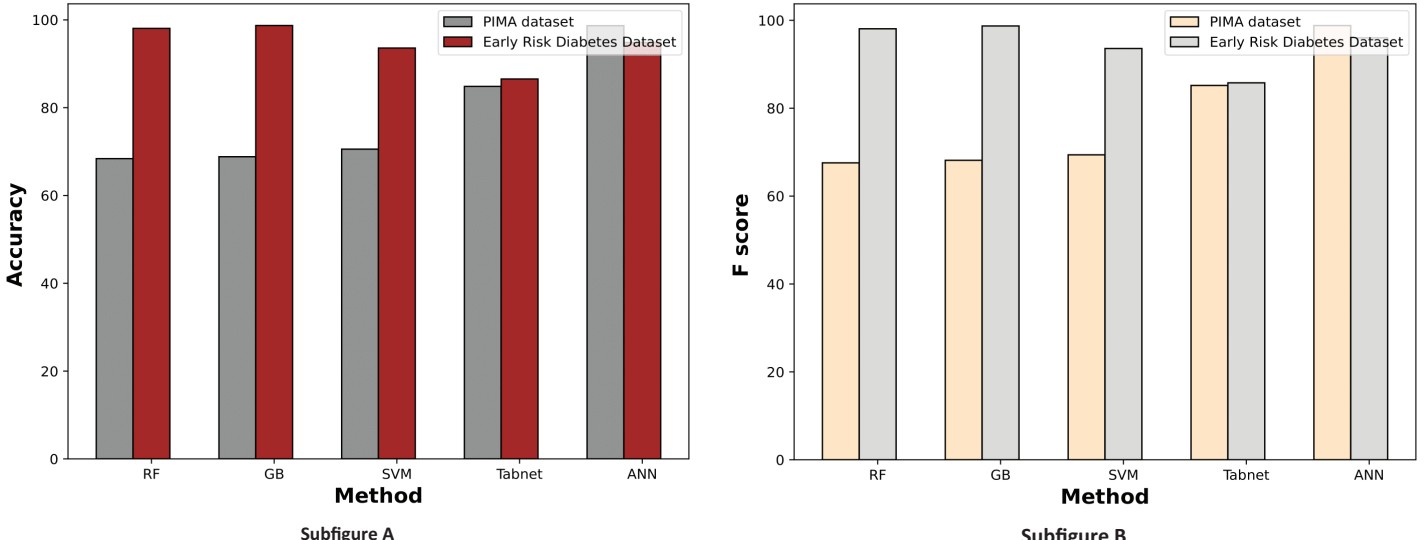

**Figure 9 Comparison of machine learning methods based on an accuracy and F score (with chi-square feature selection).**

**Table 8 Experimental results of mutual information feature selection.**

| Method | Accuracy | Precision | Recall | F score |
|---|---|---|---|---|
| Experimental results of the PIMA dataset | | | | |
| RF | 70.13% | 70.18% | 70.12% | 69.12% |
| GB | 70.56% | 70.45% | 70.6% | 69.83% |
| SVM | 71.86% | 72.09% | 71.86% | 70.91% |
| TabNet | 87.179% | 88.41% | 87.179% | 86.732% |
| ANN | 98.72% | 98.75% | 98.78% | 98.79% |
| Experimental results of the early diabetes risk dataset | | | | |
| RF | 98.72% | 98.72% | 98.72% | 98.72% |
| GB | 96.79% | 96.81% | 96.78% | 96.8% |
| SVM | 92.95% | 92.93% | 92.94% | 92.93% |
| TabNet | 90.476% | 90.51% | 90.47% | 90.488% |
| ANN | 97.44% | 96.94% | 98.96% | 98% |

## Experimental results of sequential feature selection

Machine learning methods are applied to the most relevant features, and the performance of each model is evaluated. Table 9 presents the prediction results for both datasets using RF, GB, ANN, TabNet, and SVM. The ANN classifier detects diabetes patients with 99.35% accuracy on the PIMA dataset, and the RF classifier identifies early diabetes patients with 99.36% accuracy, as shown in Table 9. The prediction performance of all other models is improved with this feature selection method on both datasets. The

**Peer**J Computer Science

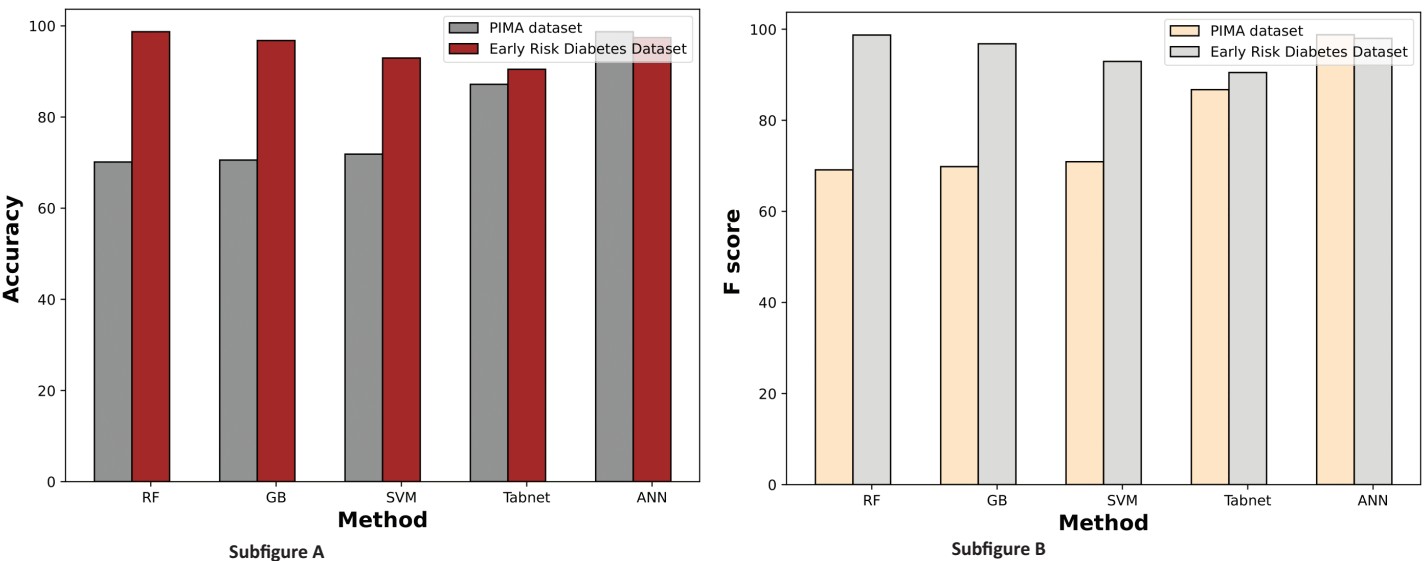

**Figure 10 Comparison of machine learning methods based on an accuracy and F score (with mutual information feature selection).**

**Table 9 Experimental results of sequential feature selection.**

| Method | Accuracy | Precision | Recall | F score |
|---|---|---|---|---|
| Experimental results of the PIMA dataset | | | | |
| RF | 74.46% | 74.42% | 74.44% | 74.01% |
| GB | 71.43% | 71.22% | 71.42% | 70.97% |
| SVM | 73.16% | 73.77% | 73.2% | 72.1% |
| TabNet | 94.871% | 95.2740% | 94.8717% | 94.799% |
| ANN | 99.35% | 99% | 99% | 99% |
| Experimental results of the early diabetes risk dataset | | | | |
| RF | 99.36% | 99.37% | 99.36% | 99.36% |
| GB | 98.08% | 98.1% | 98.077% | 98.07% |
| SVM | 93.59% | 93.59% | 93.59% | 93.59% |
| TabNet | 95.670% | 96.186% | 95.670% | 95.736% |
| ANN | 98.9% | 99% | 99% | 99% |

intuition behind this improvement is that SFS identified more informative features, enhancing the classification results of machine learning models.

Figure 11 illustrates the comparison graphs of machine learning methods based on accuracy and F-score measures for the PIMA and Early Risk Diabetes datasets. It indicates that the features selected by SFS exhibit better prediction performance for all methods than all feature sets. It suggests that selecting relevant features enhances prediction performance. This intuition arises from the fact that the selected features have a strong relationship with the target class label, and reducing the number of features helps diminish

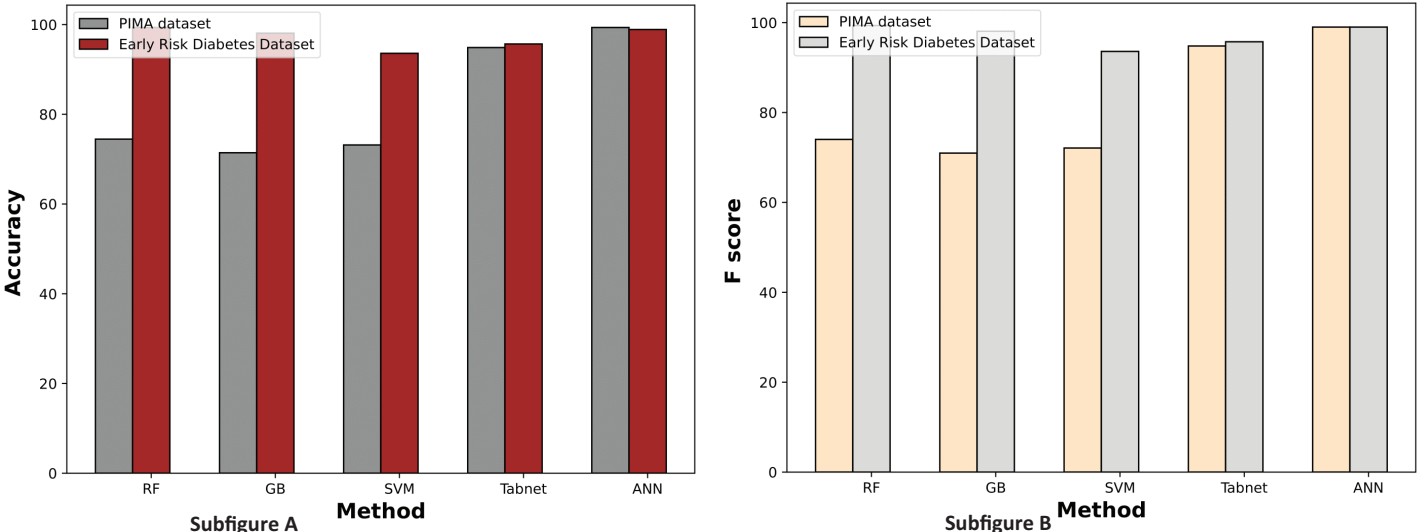

**Figure 11 Comparison of machine learning methods based on an accuracy and F score (with sequential feature selection).**

noise in the data. The results highlight that ANN and RF outperform other methods on the PIMA and Early Risk Diabetes datasets.

The confusion matrices of the machine learning models for both datasets are presented in Fig. 12 with the sequential feature selection. It demonstrates that the machine learning models have a lower misclassification error on both datasets with the sequential feature selection than without feature selection. The confusion matrices of machine learning models with the chi-square and mutual information feature selection are presented in Figs. 13 and 14, respectively.

## Comparison of the proposed method with the existing methods

### Results comparison of our proposed method with existing method on Pima dataset diabetes prediction

Many methods in the literature were applied to classify diabetes on the PIMA dataset; however, most exhibited lower prediction performance. Table 10 presents the performance of the proposed approach compared to other researchers' work on the PIMA dataset for diabetes prediction. The proposed work demonstrated an improved performance of 99.35%, surpassing existing approaches.

Table 10 concludes that the proposed method has significantly improved prediction performance on the PIMA dataset, particularly with the ANN model, compared to existing methods. Despite the numerous deep learning methods in the literature, their prediction performance falls short of the proposed approach. The selected features are crucial in reducing noise, enabling the ANN to learn a decision boundary accurately, further enhanced by hyperparameter tuning using grid search. These findings highlight the strength and efficacy of our proposed method over existing approaches.

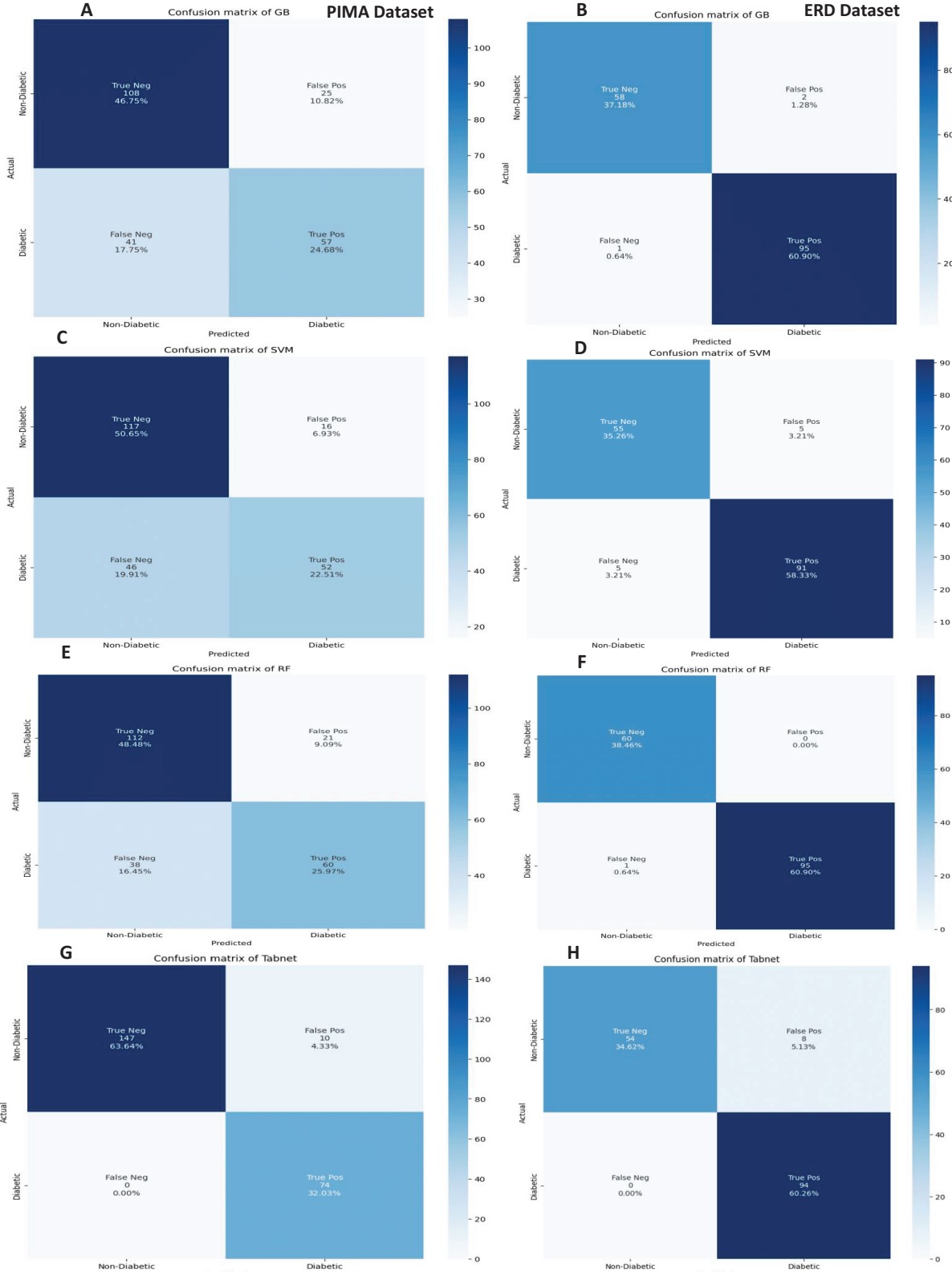

**Figure 12 Confusion metrics of machine learning models with the sequential feature selection.**

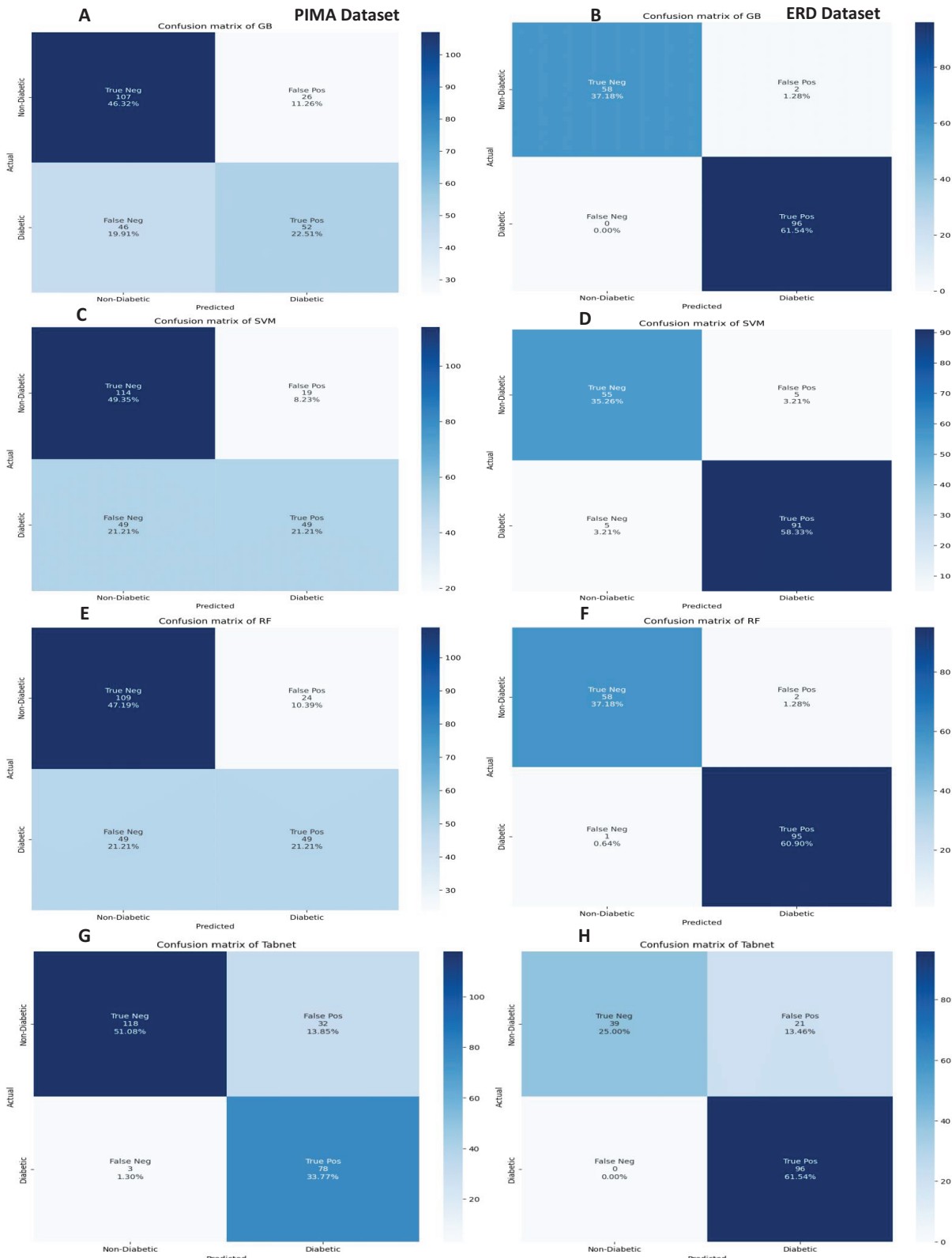

**Figure 13 Confusion metrics of machine learning models with chi square feature selection.**

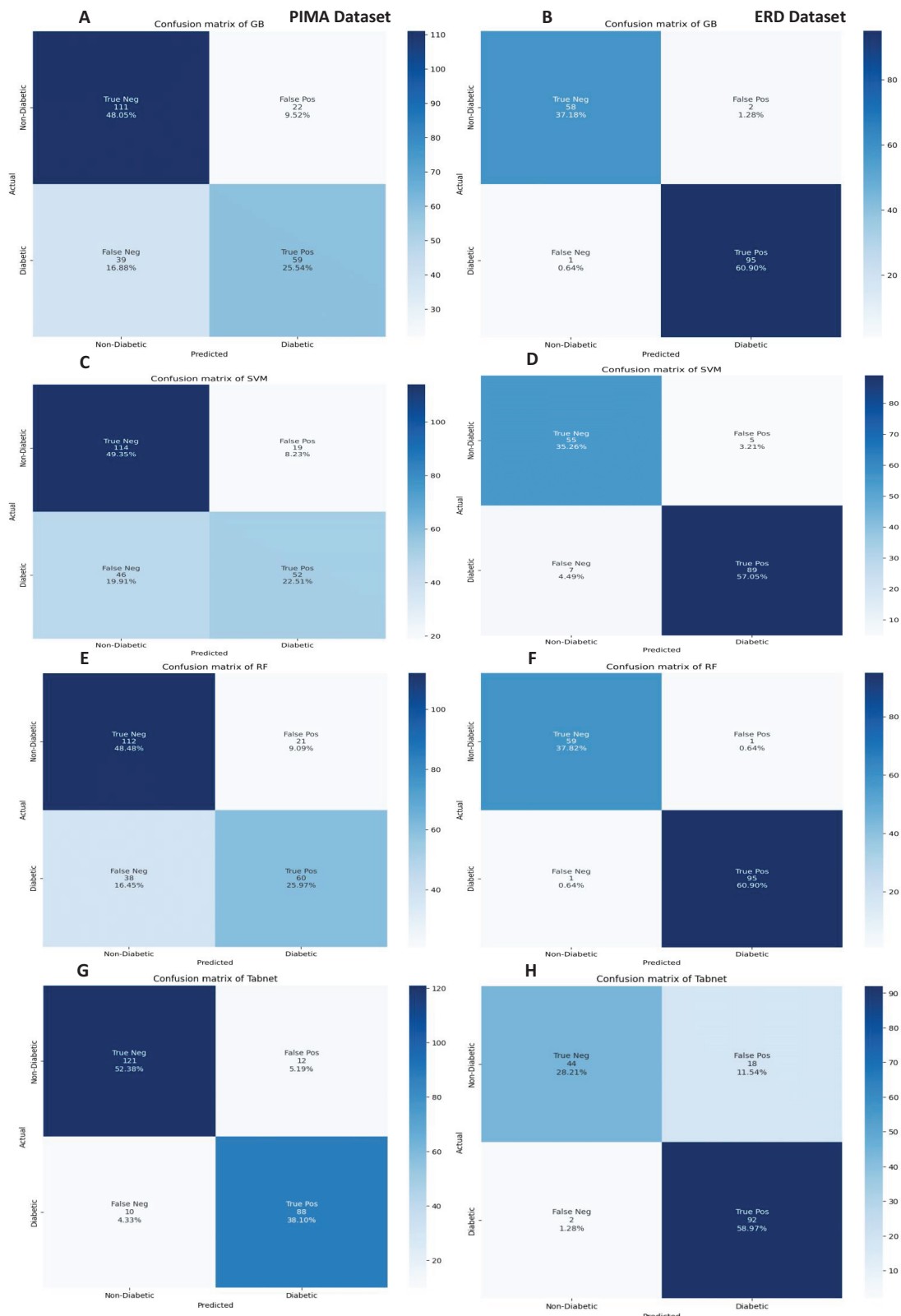

**Figure 14 Confusion metrics of machine learning models with the mutual information feature selection.**

**Table 10 Performance comparison of our work with existing approaches (PIMA dataset).**

| Method | Accuracy |
|---|---|
| Fuzzy-KNN (*Haritha, Babu & Sammulal, 2018*) | 80.3% |
| DNN+ autoencoder (*Kannadasan, Edla & Kuppili, 2019*) | 86.26% |
| DNN (*Naz & Ahuja, 2020*) | 98.07% |
| SVM (*Arora et al., 2022*) | 98.48% |
| SMOTE-SMO (*Naz & Ahuja, 2022*) | 99.07% |
| Proposed model | 99.35% |

**Table 11 Performance comparison of our work with existing approaches (ERD dataset).**

| Method | Accuracy |
|---|---|
| RF (*Shuvo et al., 2022*) | 95% |
| Neural network (*Ma, 2020*) | 96.2% |
| RF (*Laila et al., 2022*) | 97.115% |
| GA Stacking (*Tan et al., 2022*) | 98.71% |
| KNN+TMGWO+SMOTE (*Arsyadani & Purwinarko, 2023*) | 98.85% |
| Proposed model | 99.36% |

**Results comparison of our proposed method with existing method on diabetes early stage data set**

The results of the existing method on the early-risk diabetes dataset are discussed in Table 11. It is evident from Table 11 that RF with SFS feature selection achieved a prediction accuracy of 99%, surpassing the existing method's accuracy.

# CONCLUSIONS AND FEATURE SCOPE

The prevention of diabetes is crucial to avoid irreversible damage to a person's health, as this illness has claimed many lives. Early detection is vital for effective treatment. This study employs machine learning techniques for diabetes detection, focusing on two datasets: PIMA and the Early Diabetes Detection dataset. We utilized feature selection methods in the initial phase and applied ANN, RF, SVM, TabNet, and gradient boosting to the PIMA dataset. Results showed that ANN detects diabetes with 99.35% accuracy. In the second phase, we applied the SFS method with ANN, RF, and GB on the Early Diabetes Detection dataset, where RF detected diabetes patients with 99.36% accuracy. The experimental findings demonstrate that the SFS method selects the most relevant features, contributing to accurate predictions. We improved PIMA diabetes prediction to 99.35% using ANN with the SFS method. This research aids physicians in practically detecting diabetes through computer-aided diagnosis systems.

Incorporating longitudinal health data, such as lifestyle factors and genetic markers, could enhance the accuracy of early diabetes prediction models. We can utilize ensemble weighted voting classifiers to enhance the performance of the diabetes prediction task.

Exploring federated learning approaches for privacy-preserving collaboration among healthcare facilities could broaden the method's reach to include a broader range of patient groups. Additionally, researching the real-time deployment of the created model within clinical settings shows potential for prompt treatments and better diabetes management.

### Funding

This work was supported by the Institute for Information and Communications Technology Promotion (IITP) (No. 2022-0-00980, Cooperative Intelligence Framework of Scene Perception for Autonomous IoT Device). This research was also supported by the Brain Pool program funded by the Ministry of Science and ICT through the National Research Foundation of Korea (2021H1D3A2A02082991). The APC was supported by the 2023 scientific promotion program funded by Jeju National University. The funders had no role in study design, data collection and analysis, decision to publish, or preparation of the manuscript.

### Grant Disclosures

The following grant information was disclosed by the authors:
Institute for Information and Communications Technology Promotion (IITP): 2022-0-00980.
Brain Pool Program funded by the Ministry of Science and ICT through the National Research Foundation of Korea: 2021H1D3A2A02082991.
2023 Scientific Promotion Program funded by Jeju National University.

### Competing Interests

The authors declare that they have no competing interests.

### Author Contributions

- Qazi Waqas Khan performed the experiments, performed the computation work, prepared figures and/or tables, and approved the final draft.
- Khalid Iqbal conceived and designed the experiments, performed the experiments, authored or reviewed drafts of the article, and approved the final draft.
- Rashid Ahmad conceived and designed the experiments, performed the experiments, authored or reviewed drafts of the article, and approved the final draft.
- Atif Rizwan analyzed the data, performed the computation work, prepared figures and/or tables, and approved the final draft.
- Anam Nawaz Khan analyzed the data, performed the computation work, prepared figures and/or tables, and approved the final draft.
- DoHyeun Kim conceived and designed the experiments, authored or reviewed drafts of the article, and approved the final draft.

### Data Availability

The raw measurements are available in the Supplemental Files.

The Early Stage Diabetes Risk dataset is available at: https://archive.ics.uci.edu/ml/datasets/Early+stage+diabetes+risk+prediction+dataset.

The PIMA dataset is available at: https://networkrepository.com/pima-indians-diabetes.php.

## Supplemental Information

Supplemental information for this article can be found online at http://dx.doi.org/10.7717/peerj-cs.1914#supplemental-information.

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
