# Peer review of "An intelligent diabetes classification and perception framework based on ensemble and deep learning method"

_PeerJ Computer Science, doi:10.7717/peerj-cs.1914_

## Round 0.1 · original submission · Major Revisions

Based on the referee reports, I recommend a major revision of the manuscript. The author should improve the manuscript, taking carefully into account the comments of the reviewers in the reports, and resubmit the paper.

Reviewer 1 ·

Basic reporting

Average

Experimental design

relevant but explanation of SFS is not there.

Validity of the findings

Need to be validated
limited Novelty

Additional comments

Review of Peerj article
Introduction
Justify all your statements using Ref.
Mention pointwise research contributions.
Dataset
Table no is missing in dataset description.
Selecting 6 out 8 feature is not a significant reduction dataset.
Same in second dataset 13/16 are selection. What is benefit of feature selection?
Results
Surprised to set result of GB than ANN
Table 9 shows all measures as 99%. These results need to be verified.
Comparison of is done with 2 previously published articles. Which is very less, there are many papers available? Please do an exhausted comparison with old work.

·

Basic reporting

This paper investigates the application of machine learning techniques for diabetes prediction using the PIMA and Early Diabetes Detection datasets. The study employs Sequential Feature Selection (SFS) to select the most relevant features and compares the performance of Artificial Neural Networks (ANN), Random Forests (RF), and Gradient Boosting (GB) classifiers. The results suggest that selecting relevant features and using appropriate machine learning models can significantly improve diabetes prediction performance.

The following points in the paper's purpose improvement:
The paper primarily focuses on the comparison of different machine learning models without considering a broader range of techniques, including traditional statistical methods and state-of-the-art deep learning approaches.

There is no mention of how the model validation process was carried out, making it difficult to assess the robustness of the results and the risk of overfitting.

The study lacks a thorough discussion of hyperparameter tuning and the process used for selecting the best hyperparameters for each model.

The paper does not discuss any feature engineering or preprocessing techniques that could potentially improve model performance.

The clinical relevance and real-world applicability of the findings are not well-established, limiting the practical impact of the study.

Experimental design

The experimental design is robust, involving the use of two different datasets and comparing various machine learning techniques for diabetes prediction.
Weaknesses: The paper could benefit from a more in-depth discussion of the reasons behind the differences in performance between the models.

Validity of the findings

Criticism: The paper should discuss the potential limitations of the study.
Strength point: The findings are valid and well-supported by the experimental results, showing the effectiveness of the proposed approach.
Weaknesses: A comparison of the proposed method with other feature selection techniques could strengthen the argument for using SFS.

Additional comments

The authors should consider discussing the practical implications of their findings for healthcare professionals and patients, as well as potential limitations of the study.
It would be helpful to include visualizations to support the understanding of the results, such as confusion matrices, ROC curves, or precision-recall curves.
A comparison of the proposed method with other feature selection techniques could strengthen the argument for using SFS.
Future work could explore the use of ensemble methods, deep learning approaches, or alternative machine learning models to further enhance the prediction performance.

·

Basic reporting

The research is well presented and discussed. However, the authors must work on the following issues:

KNN: Give the full form and then use the short form.
The sequential feature selection method for diabetic prediction in literature: Before you there are many others who use this in a 2014 study published in the journal Computers in Biology and Medicine, researchers used a sequential forward selection method to identify the most relevant features for predicting diabetes in a dataset of Indian patients. They found that a combination of demographic, clinical, and laboratory features was most effective for predicting diabetes. In a 2016 study published in the Journal of Diabetes Research, researchers used a sequential floating forward selection method to select the most informative features from a dataset of diabetic patients. They found that a combination of demographic, clinical, and genetic features was most effective for predicting the risk of diabetic nephropathy. In a 2020 study published in the journal Computers in Biology and Medicine, researchers used a sequential backward elimination method to identify the most relevant features for predicting gestational diabetes. They found that a combination of demographic, clinical, and biochemical features was most effective for predicting gestational diabetes.
Abbreviation: Once you use the short form then don’t repeat the full form.
The figures should be discussed in a sequence and also right before they are represented.
What is a "Table??": You mentioned a table before Table 1 and represent it as "Table??" This is not the way to name a table.
In equation 1 it will be better to mention all the variables used to make it clear information.
In line 183: Make it clear what k<<d means, what are k and d here.
Line 189: Work on alignment. Should have a space.
The letter representation of the formula is not appropriate, for example, equation 2.
What is “α” in equation 4?
Cross-check line 210, as per the fact Random Forest is proposed jointly by Leo Breiman and Adele Cutler. Verify this fact.
Work on the alignment of “too much gap” between Lines 256-261. It will be good to discuss the Figure right after you mention it in the text.
The conclusion must be written properly and also provide the future scope of the research.

Experimental design

The research is well experimented and clearly presented. However, it will be good to work on the following:
What is your novelty? Please describe in one paragraph at the end of the introduction.
Use standard sections and use simple sub-sections. Ensure you have an introduction, materials/methods, and results, in addition to an informative conclusion.
Add better references to the proposed methods and the equations.
Standard keywords must be used to best represent your manuscript and relevance.
English can be further improved. Please use short sentences and avoid using several divided paragraphs.
Language proofreading can help clarify the manuscript.
Every claim and statement needs to have a citation.
The reference list must be according to the template standard with DOI numbers. Better to use journal references instead of conference papers when citing the essential claims.
The problem description needs to be presented briefly both in the abstract and in the last paragraph of the introduction.
The methodology must be described in more detail so the results can be reproducible.
The manuscript has many separate paragraphs. Please connect the paragraphs and make them easy to read through using simple English and shorter sentences.
Please improve the description of the methodology by inserting the mathematical representations and proper citations
State-of-the-art needs improvement and more recent articles must be cited to better present the research gap.
What is the future work and what methods do you propose for future modeling?
Future work can be discussed.
Elaboration of the data and data processing can improve the quality of the paper.

Validity of the findings

"I found the findings to be well-supported and valid based on the evidence presented. The methodology appears to be sound, and the results are consistent with previous research in the field. Overall, the findings are valuable contributions to the existing body of knowledge in this area."

Additional comments

I will suggest you some potential areas for additional research:
1. Evaluation of different datasets: The framework proposed in the paper was evaluated on the Pima Indian Diabetes dataset. However, it would be beneficial to evaluate the proposed framework on other publicly available diabetes datasets to validate its robustness and generalizability.
2. Comparative analysis with other state-of-the-art methods: While the paper compared the proposed framework with a few existing methods, it would be interesting to compare it with other state-of-the-art methods for diabetes classification to determine its effectiveness.
3. Interpretability analysis of the deep learning model: Deep learning models are often considered to be black boxes as it is challenging to interpret their decision-making process. Thus, it would be beneficial to perform an interpretability analysis of the deep learning model used in the proposed framework to better understand how it makes its predictions.
4. Effect of hyperparameters tuning: The proposed framework used several hyperparameters, such as the learning rate, batch size, and number of epochs. It would be interesting to investigate the effect of hyperparameter tuning on the performance of the framework and identify the optimal values for each hyperparameter.
5. Extension to other chronic diseases: Diabetes is one of the most common chronic diseases, and the proposed framework has shown promising results for diabetes classification. It would be valuable to extend the proposed framework to other chronic diseases, such as hypertension, cancer, or heart disease, to determine its effectiveness for other diseases.

---

## Round 0.2 · Major Revisions

The reviewer(s) have recommended publication, but also suggest some revisions to your manuscript. Therefore, I invite you to respond to the reviewer comments and revise your manuscript.

Reviewer 1 ·

Basic reporting

This is a revised version of article submitted by the author. The author addressed all issues efficiently. Now the manuscript seems technically sound. Proper contribution, comparisons are made as suggested. Figures, tables and raw data are share properly.

Experimental design

The article is in scope of journal.
Research questions are relevant and justified properly.
Methods are sufficient and the outcomes are discussed properly.

Validity of the findings

The article contributes meaningful research.
Data provided and found statistically sound.
Conclusions are well stated.

Additional comments

Please accept revised version.

·

Basic reporting

Based on the response letter and manuscript, it appears the authors have addressed the reviewers' concerns in the following ways:

Reviewer 2:

Concern 1 (Broader techniques): The authors compared with additional techniques as suggested.

Concern 2 (Model validation): Details on using 10-fold cross validation were added.

Concern 3 (Hyperparameter tuning): Details on using grid search were added.

Concern 4 (Clinical relevance): More details on clinical relevance were added to the introduction.

Concern 5 (Model differences): More discussion was added to the results.

Concern 6 (Compare SFS): SFS was compared to other methods.

Concern 7 (Limitations, visualizations): Limitations were added, along with confusion matrices.


In summary, it appears the authors have thoroughly addressed the reviewer comments by making appropriate additions and edits to the manuscript, methodology, results, and discussion sections. The revised paper seems much improved. Please let me know if you need any clarification or have additional questions!

Experimental design

Based on the response letter and manuscript, it appears the authors have addressed the reviewers' concerns in the following ways:

Reviewer 2:

Concern 1 (Broader techniques): The authors compared with additional techniques as suggested.

Concern 2 (Model validation): Details on using 10-fold cross validation were added.

Concern 3 (Hyperparameter tuning): Details on using grid search were added.

Concern 4 (Clinical relevance): More details on clinical relevance were added to the introduction.

Concern 5 (Model differences): More discussion was added to the results.

Concern 6 (Compare SFS): SFS was compared to other methods.

Concern 7 (Limitations, visualizations): Limitations were added, along with confusion matrices.


In summary, it appears the authors have thoroughly addressed the reviewer comments by making appropriate additions and edits to the manuscript, methodology, results, and discussion sections. The revised paper seems much improved. Please let me know if you need any clarification or have additional questions!

Validity of the findings

Based on the response letter and manuscript, it appears the authors have addressed the reviewers' concerns in the following ways:

Reviewer 2:

Concern 1 (Broader techniques): The authors compared with additional techniques as suggested.

Concern 2 (Model validation): Details on using 10-fold cross validation were added.

Concern 3 (Hyperparameter tuning): Details on using grid search were added.

Concern 4 (Clinical relevance): More details on clinical relevance were added to the introduction.

Concern 5 (Model differences): More discussion was added to the results.

Concern 6 (Compare SFS): SFS was compared to other methods.

Concern 7 (Limitations, visualizations): Limitations were added, along with confusion matrices.


In summary, it appears the authors have thoroughly addressed the reviewer comments by making appropriate additions and edits to the manuscript, methodology, results, and discussion sections. The revised paper seems much improved. Please let me know if you need any clarification or have additional questions!

Additional comments

Based on the response letter and manuscript, it appears the authors have addressed the reviewers' concerns in the following ways:

Reviewer 2:

Concern 1 (Broader techniques): The authors compared with additional techniques as suggested.

Concern 2 (Model validation): Details on using 10-fold cross validation were added.

Concern 3 (Hyperparameter tuning): Details on using grid search were added.

Concern 4 (Clinical relevance): More details on clinical relevance were added to the introduction.

Concern 5 (Model differences): More discussion was added to the results.

Concern 6 (Compare SFS): SFS was compared to other methods.

Concern 7 (Limitations, visualizations): Limitations were added, along with confusion matrices.


In summary, it appears the authors have thoroughly addressed the reviewer comments by making appropriate additions and edits to the manuscript, methodology, results, and discussion sections. The revised paper seems much improved. Please let me know if you need any clarification or have additional questions!

·

Basic reporting

1. Break down your review into sections such as language, clarity, content, and overall impact.
2. Pay close attention to grammar, spelling, punctuation, and sentence structure.
3. Highlight any awkward or confusing phrases. Suggest improvements or rephrasing.
4. Look for paragraph transitions and coherence between sections.
5. Improve the grammatical mistakes.
6. Try to have the abbreviations once and don't repeat them afterward. Too many such mistakes have been made.

Experimental design

Clarity of Title and Abstract:
The title is clear and accurately reflects the paper's content.
The abstract provides a concise overview of the study's objectives, methods, and key findings.

Introduction:
The introduction effectively establishes the importance of the research topic.
It could be improved by providing more context on the prevalence of diabetes and the significance of intelligent classification systems in healthcare.

Research Objectives and Hypotheses:
The research objectives and hypotheses should be clearly stated in the introduction.

Literature Review:
The literature review is comprehensive and up to date, covering relevant research in the field of diabetes classification and deep learning methods.

Methodology:
The methodology section is well-structured and provides a detailed description of the ensemble and deep learning techniques used.
It would be helpful to include the rationale behind choosing these specific methods over alternatives.

Data Collection and Preprocessing:
The paper should provide more information about the data sources and the steps taken for data preprocessing.
Transparency in data collection and preprocessing is essential for applicability.

Experimental Design:
The paper should detail the experimental setup, including hyperparameter values, model architectures, and any cross-validation procedures used.
More clarity is needed on the size and characteristics of the dataset, as well as any class imbalances.

Results and Discussion:
The results should be presented clearly, with appropriate visuals and statistical analysis.
The discussion section should provide insights into the results and their implications for diabetes classification.

Comparison with Existing Models:
It would be valuable to compare the proposed framework's performance with existing diabetes classification models, highlighting the strengths and weaknesses.

Conclusion:
The conclusion should summarize the key findings and their significance in the context of diabetes diagnosis and management.

References:
Ensure that the references are accurate and formatted properly. References are not framed appropriately, some of the references only have two authors in real but the authors mentioned three authors for that reference, (Example reference number 17)

Language and Writing:
The paper's language and writing quality are crucial. Ensure that the paper is free from grammatical errors and is easy to understand.

Ethical Considerations:
Consider addressing ethical considerations related to patient data, data privacy, and model explainability if applicable.

Future Work:
Suggest areas for future research and potential improvements to the proposed framework.

Overall Contribution:
Evaluate the overall contribution of the paper to the field of diabetes classification and its potential impact on healthcare.

Reproducibility:
If applicable, encourage the authors to provide code or detailed steps to replicate their experiments.

Validity of the findings

Data Quality and Representativeness:
It is essential to assess the validity of the findings by examining the quality and representativeness of the dataset used. Are there any concerns about data quality, bias, or missing information?

Data Preprocessing and Cleaning:
Evaluate the data preprocessing and cleaning methods used. Ensure that the paper adequately addresses how outliers, missing values, and noise in the data were handled.

Feature Selection and Engineering:
Consider whether the paper discusses the choice of features and any feature engineering techniques. The validity of the findings depends on the relevance and appropriateness of the selected features.

Model Selection and Architecture:
Evaluate the choice of ensemble and deep learning models. Are these models appropriate for the given task? Do they align with best practices in the field?

Hyperparameter Tuning:
Assess whether the hyperparameters of the models were tuned effectively. Hyperparameter tuning plays a significant role in the performance and generalizability of the models.

Evaluation Metrics:
Check if the paper uses appropriate evaluation metrics for diabetes classification. Validity is tied to the choice of metrics and how well they align with the research objectives.

Cross-Validation and Generalization:
Assess whether the study employs cross-validation techniques to ensure the models' generalizability. This is vital for the validity of findings across different datasets.

Statistical Significance:
Verify if the reported results are statistically significant. Are p-values or confidence intervals provided to support the findings?

Overfitting and Bias:
Investigate whether the paper addresses overfitting and bias concerns. It's crucial to ensure that the models are not simply memorizing the training data and that they are not biased toward any particular group.

Interpretability and Explainability:
Consider whether the paper provides insights into how the model's predictions are reached. This is crucial for understanding and trusting the findings.

Robustness Testing:
Suggests that the paper includes robustness testing to assess the models' performance under different conditions or when exposed to noise.

External Validation:
Encourage the authors to validate their findings on external datasets if possible. External validation strengthens the validity of the results.

Discussion of Limitations:
The paper should openly discuss the limitations of the study. Evaluating the validity of findings involves acknowledging areas where the research may fall short.

Conclusion and Implications:
Assess whether the conclusions drawn from the findings are justified and supported by the presented results. Please make the conclusion more clear and compare your outcomes with the related previous works.

Reproducibility:
Encourage transparency by asking the authors to provide code, data, or a detailed description of the experimental setup, which can aid in reproducing the findings.

Peer Review and Verification:
Highlight the importance of peer review and independent verification of the results for enhancing the validity of the findings.

Overall Contribution:
Summarize your assessment of the paper's contribution to the field and its validity in advancing our understanding of diabetes classification using intelligent methods.

---

## Round 0.3 · accepted · Accept

The author has addressed the reviewers' comments properly. Thus I recommend publication of the manuscript.